# Sharp Profile for Icebreaking Propellers to Improve Their Ice and Hydrodynamic Characteristics

**Alexander Vladislavovich Andryushin** [1,*]**, Sergey Vladimirovich Ryabushkin** [2]**, Alexander Yurievich Voronin** [2] **and Egor Vladimirovich Shapkov** [2]

[1]    Laboratory of Propulsion Complexes of Ships, Central Research and Design Institute (CNIIMF), 191015 Saint-Petersburg, Russia
[2]    Central Research and Design Institute (CNIIMF), 191015 Saint-Petersburg, Russia; ryabushkinsv@cniimf.ru (S.V.R.); voroninay@cniimf.ru (A.Y.V.); shapkovev@cniimf.ru (E.V.S.)
*    Correspondence: propulsionlab@cniimf.ru; Tel.: +7-(812)-271-12-83

**Abstract:** Stern-first operation under severe ice conditions (ridges) is one of the most effective modes to increase the operating efficiency of icebreakers and ice ships. However, when a ship overcomes the ridge astern, the propellers continuously interact with ice blocks, and ice moment affects the propeller and the main engine. This leads to propeller speed drop and propeller thrust reduction. Propeller stop is also possible. This is the reason why the propeller ice moment needs to be decreased. Blade profiles with a sharp leading edge are used for this purpose because their thickness is significantly less than that of a traditional icebreaking profile. The application of sharp profiling makes it possible to significantly reduce the ice moment (ice loads) on the propeller, reduce the drop in its speed, and increase the hydrodynamic thrust. The main task when installing blades with sharp profiles is to ensure the strength of their leading edges exposed to ice pressure. In this article, the authors tackle upon some methods of assigning integral and local ice loads on propellers. Solutions for ensuring the local strength of the blade edges were developed and presented. The influence of sharp profiling on the hydrodynamic and cavitation characteristics of ice propellers was considered. The article presents examples of calculating the hydrodynamic propeller thrust and moment, as well as ice loads on a propeller with a sharp and traditional profile, when an ice ship moves through a ridged ice isthmus with its stern first.

**Keywords:** icebreaker propeller; sharp profile; strength of propellers; propeller/ice interaction; hydrodynamics; cavitation

## 1. Setting Up an Issue: The Main Approaches to Propeller Design for Modern Ice-Going Vessels

Ensuring the strength and performance of propulsion systems, as well as increasing their operational efficiencies, are some of the knotty tasks in the design of modern Arctic ships.

Nowadays, these tasks are brought into focus by the intensive development of oil, gas, and other fields on the Arctic shelf of Russia. Efficient transportation of a large volume of hydrocarbons and other cargoes requires an increase in the number of icebreaking transport vessels and their cargo capacities, support icebreakers, and extension of navigation periods in ice conditions. The use of the reverse mode (propellers forward) is a well-known way to significantly increase the ice propulsion and operational capability of icebreakers and ice-going vessels in ice. This principle (propellers forward) is the basis of the double-acting ship (DAS) concept.

Propellers-forward movement of a DAS is the main operating mode in heavy ice conditions. This significantly improves the propulsion of the vessel (speed) moving independently and under the assistance of icebreakers in heavily ridged ice, reduces the cost of icebreaking support, and increases the efficiency of cargo transportation (gas, oil, containers).

Construction of large-capacity tankers and the LNGC double-acting ships "Shturman Albanov" and "Christophe de Margerie" of the Arctic class Arc7 made it possible to implement projects for the year-round export of oil and LNG from the Gulf of Ob. In winter–spring navigation, transportation is carried out in the Western direction through the Kara Sea (Western sector of the NSR), and in summer–autumn, navigation to South Asia through the eastern sector of the NSR. It is important to point out that the increased power and astern capability to break 2 m thick ice ensure the independent operation of these vessels in the Western sector of the NSR and the efficiency of LNG transportation in a context of a deficit of modern Arctic icebreakers.

Novatek's new Arctic LNG 2 project is the next stage in the development of Arctic oil and gas fields. As part of this project, it is planned to transport LNG year-round by large-capacity Arctic gas carriers (DAS) to South Asia along the eastern sector of the NSR with a trans-shipment terminal (hub) in Kamchatka. The task is complicated by the lack of the required number of Arctic icebreakers to provide traditional icebreaking assistance (in the channel behind the icebreaker). In this case, icebreaking support is carried out in the most difficult areas of the route with multiyear ice (the Laptev and East Siberian seas). The areas with lighter ice conditions are overcome by the vessels independently. For the efficient functioning of such a transport system, the ice capability of prospective LNGCs moving astern should be at least 2.0 m; this demands a propeller power exceeding $Np > 10$ MW.

In the light of the foregoing, a common trend of modern Arctic shipping is the development and operation of large-capacity tankers and LNGCs of the DAS type with increased power and ice capability. Such vessels with propeller power $Np > 10$ MW are equipped with electric azimuth thrusters supplied by ABB or GE.

Ensuring the strength of azimuth thrusters and icebreaking propellers, as well as the operability of the main electric engine (MEE) for double-acting ships, is one of the key tasks of their design due to the high level of ice loads on the bow azimuth thrusters (propellers). Ice strengthening of the main elements of the propulsion line in the flow of power lines, including propellers, is assigned due to the condition of both fatigue strength assurance and strength from one-time extreme ice loading.

The operability of an MEE is understood as its ability to withstand the ice moment; to maintain the specified power, torque, and speed of the propeller to ensure sufficient thrust and movement of the vessel; and to prevent its stop and possible blade damage. The effect of the ice moment on the propeller and MEE leads to a decrease in the speed of the propeller and a drop in the thrust of the propulsion complex. During prolonged interactions between the propeller and ice, when the DAS vessel overcomes the ridge astern (propellers forward), these processes significantly reduce the propulsion of the vessel and may lead to a stop. The impact of an ice moment on the MEE that exceeds its design value can lead to propeller stop and blade damage [1]. An increase in the MEE design moment for modern azimuth thrusters is associated with an increase in its diameter and the diameter of the pod, which leads to a decrease in its propulsion efficiency. Therefore, it is extremely important to reduce the ice moment on the propellers to ensure the operability of an MEE and to realize the advantages of the astern movement in ice conditions. The latter is confirmed by the experience of designing and operating Arctic icebreakers and large-capacity Arctic tankers ("Vasily Dinkov", "Shturman Albanov", "Christophe de Margerie"). Application of sharp profiles (profiles with sharp leading edges) is one of the most effective solutions for reducing the level of ice moments on the propellers, increasing the operational thrust of the azimuth unit, the operability of the MEE, and the operational efficiency of the ship moving astern. The use of a sharp profile also significantly improves the hydrodynamic and cavitation characteristics of the propeller. However, the widespread use of sharp profiles was limited by ensuring the strength of the sharp (thin) blade leading edges under ice loading. These aspects of the design of icebreaking propellers with sharp profiles are discussed below.

## 2. Traditional (Blunt) and Sharp Profiles for Icebreaking Propellers: Main Mechanisms of Ice Breaking by Blade Edges

At present, in Russian practice, for the design of propellers for ships intended for operation in open water and in light ice conditions (without ice classes and with low ice strengthening), profiles of the NACA 66-mod type are widely used. For icebreakers and ships of high Arctic classes, the IK82 profile developed by KSRC [2,3] is used. Figure 1 shows the thickness distribution for these profiles in the area of the leading edge (profile tip). Figure 1 also shows the profile shapes. The IK82 profile has thicker leading (trailing) edges compared to the NACA 66-mod profile, which is due to the need to ensure their strength under the influence of ice loading.

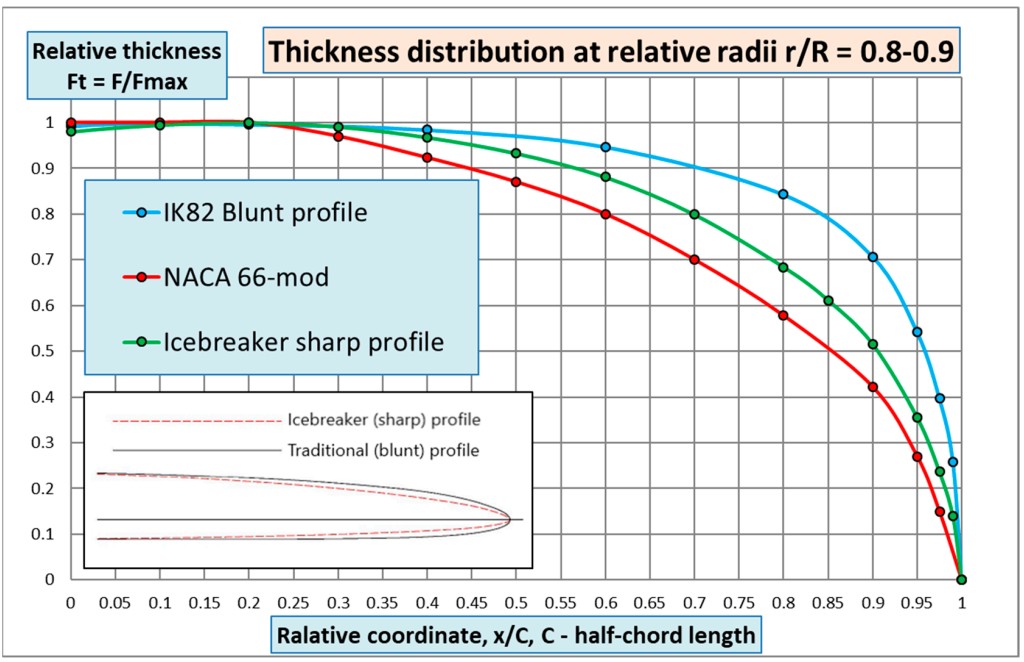

**Figure 1.** Thickness distribution for profiles of type IK82 (blunt profile), NACA 66-mod, and icebreaking profile.

An increase in the thickness of the leading edge significantly reduces the hydrodynamic and especially the cavitation characteristics of propellers [2]. It should be pointed out that the thickness distribution for the IK82 profile is empirical; i.e., it was obtained on the basis of operating experience without a calculation substantiation of the strength from the effect of ice loading. This approach did not give the designer the opportunity to reasonably reduce the thickness of the profile to improve the performance of the propeller. In the 1980s, according to V.A. Belyashov's proposal, a sharp icebreaking profile was installed on the side propeller of the Arctic icebreaker (see Figure 1) with a thickness close to that of the NACA 66-mod profile [4]. Full-scale trials of the icebreaker showed that the use of a sharp profile made it possible to reduce the ice moment on the propeller by 25% [4]. Operating experience has shown that the sharp edges of blades made of high-strength propeller steels are not subject to destruction from ice loads. However, the use of the propellers with a sharp icebreaking profile in a wide range of diameters and disk ratios was hampered by a lack of design models of ice loads (pressures) on the edges of the blades to ensure their strength. At present, such a model can be formed on the basis of studies carried out by V.A. Belyashov, N. Soininen, B. Veitch, and A.V. Andryushin [4–7]. The main provisions and approaches are presented below. The ice-milling mode with a positive angle of attack of the profile was taken as the design mode for assigning ice loads [8]. For the specified mode, Figure 2 shows a general scheme of the destruction of ice by the edge of the propeller blade (the first type of ice milling [4–8]). The detailed ice damage scheme (Figure 3) is presented and considered below.

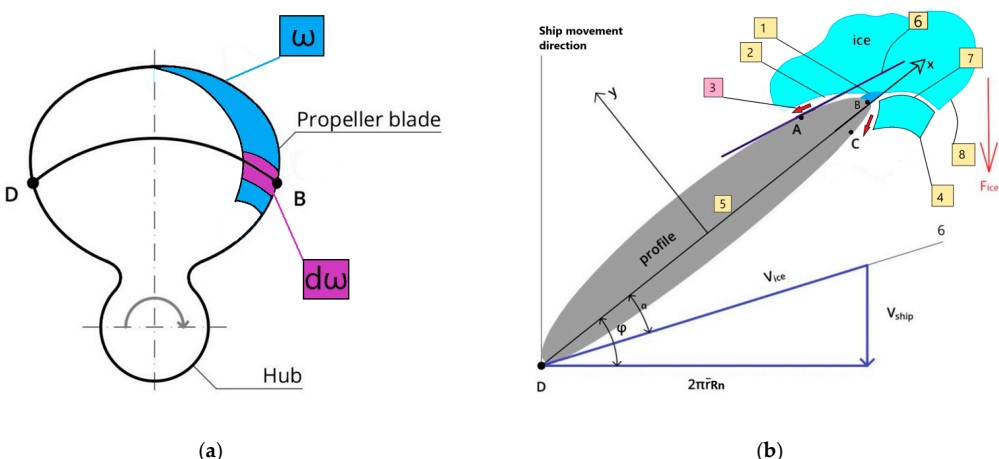

**Figure 2.** (**a**) Loaded area on the propeller blade; (**b**) The process of ice damage by a blade leading edge for ice milling at positive attack angle (V.A. Belyashov, N. Soininen, B. Veitch, Andryushin) [4–8]. DB—cylindrical section of propeller blade; AB—contact zone from the suction side; BC—contact zone from the pressure side; $\omega$—length of contact zone along leading edge; $d\omega$—element of contact zone; 1—ice crushing zone; 2—ice channel; 3—ice powder; 4—split fragment; 5—profile chord; 6—line of attack angle; 7—splitting crack; 8—ice free surface; $\alpha$—attack angle; $\varphi$—pitch angle; $V_{ship}$—ship speed; $V_{ice}$—ice contact speed; $n$—propeller speed; $\bar{r}$—relative radius of propeller; $R$—propeller radius; $F_{ice}$—backward axial ice force (see below).

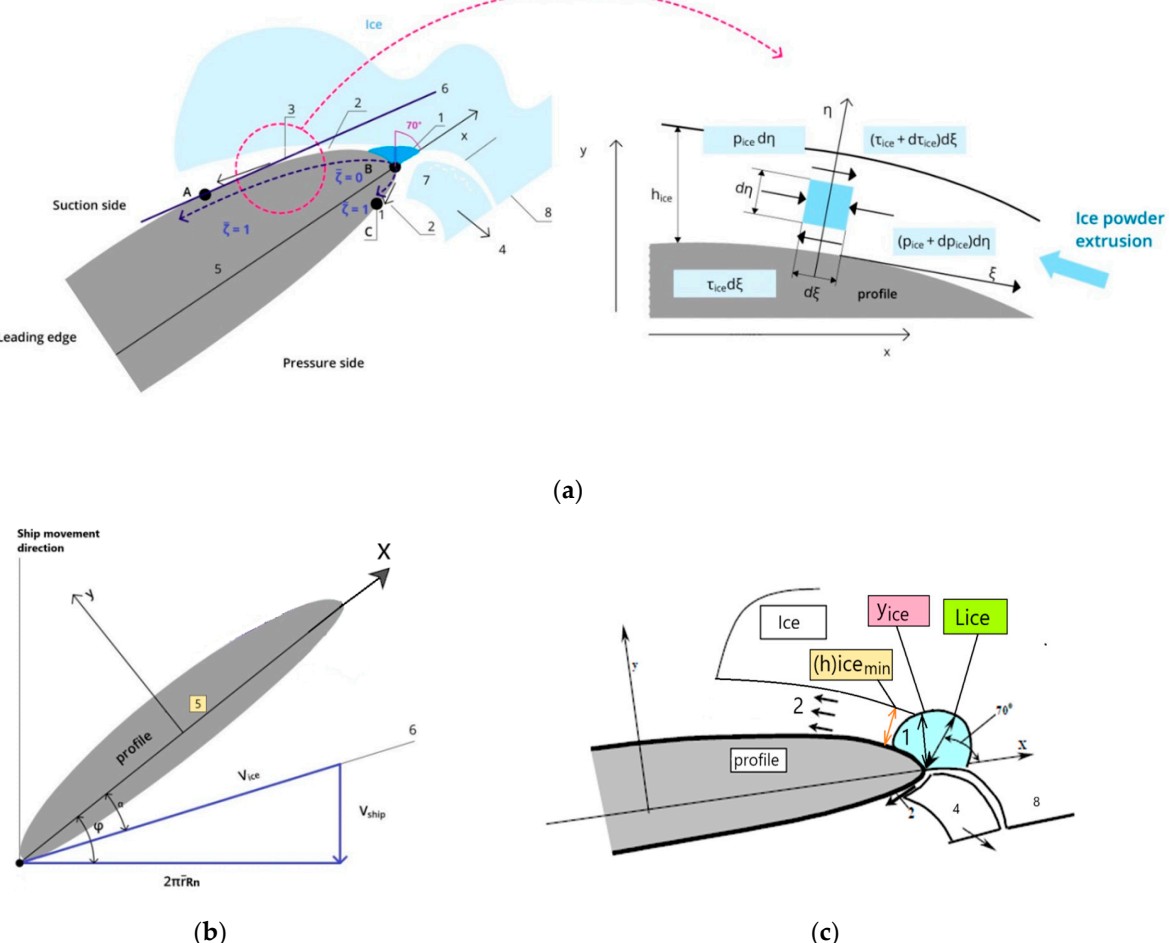

**Figure 3.** (**a**) Ice powder extrusion process and equilibrium condition of an ice powder element [5,7]; (**b**) Coordinate systems used and profile velocities during propeller/ice interaction; (**c**) Ice-crushing

zone. AB—contact zone from the suction side; BC—contact zone from the pressure side; $\overline{\zeta} \in [0, 1]$—dimensionless ice contact zone along pressure and suction sides of a profile, 0—beginning of a contact zone (point B); 1—end of a contact zone (points A, C); 1—ice crushing zone; 2—ice channel; 3—ice powder; 4—split fragment; 5—profile chord; 6—line of attack angle; 7—splitting crack; 8—ice free surface; $\alpha$—attack angle; $\varphi$—pitch angle; $V_{ship}$—ship speed; $V_{ice}$—ice contact speed; $n$—propeller speed; $\overline{r}$—relative radius of propeller; $R$—propeller radius; $l_{ice}$—typical size of an ice-crushing zone (see Equation (10)); $(h_{ice})_{min}$—typical thickness of ice-powder layer (channel width) at profile tip. The angle of $\approx 70°$ was determined using an ice linear fracture mechanics approach while assuming that the stress–strain state in ice follows the condition of plane strain (see Equation (10) for an explanation).

During interaction of the propeller blade edge with the ice, on the tip of the propeller profile, the ice is crushed under a confined stress state. The size of the ice-crushing zone 1 along a profile plane was about (5–10) mm based on experimental results [5]. These results are confirmed below for multiyear and fresh ice with low salinity (porosity) using ice-fracture-mechanics approaches. In the ice-crushing zone 1, ice pressure $(p_{ice})_{max}$ (crushing strength or compressive strength under triaxial stress state) for a narrow strip of a length $d\omega \to 0$ can be determined by using Equation (1), which was developed using experimental data [5]:

$$(p_{ice})_{max}(d\omega) = 15\sigma_{compr}{}^{0.6} \tag{1}$$

where $\sigma_{compr}$ is the unconfined compressive strength.

In practical strength calculations (see below), the value of $d\omega$ is recommended to be taken equal to the typical size of an ice sample to determine the unconfined compressive strength value.

Along the edge of the blade in the crushing zone 1 (see Figure 2), the ice is destroyed nonsimultaneously, which causes a scaling effect of the ice-crushing strength $(p_{ice})_{max}$; i.e.,:

$$(p_{ice})_{max}(\omega) = (p_{ice})_{max}(d\omega)k_{scale} = 15\sigma_{compr}{}^{0.6}\, k_{scale} \tag{2}$$

where $k_{scale}$ is the scaling factor. Calculation of the value of the $k_{scale}$ can be performed by using Equation (3) obtained from the experimental data [9]:

$$k_{scale}(l) = 0.7853 \cdot e^{-l/1.99} + 0.2146. \tag{3}$$

In the ice-crush zone, shear stresses are determined according to the Amontons–Coulomb friction law:

$$(\tau_{ice})_{max}(d\omega) = \mu_d\, (p_{ice})_{max}(d\omega);\ \ (\tau_{ice})_{max}(\omega) = \mu_d\, (p_{ice})_{max}(\omega) \tag{4}$$

where $\mu_d$ is the coefficient of dynamic friction between the blade and broken ice.

Crushed ice in the form of a powder 3 (see Figures 2 and 3) is pushed out on an expanding channel 2 along the suction and pressure surfaces of the blade profile, and then it forms a thin intermediate layer between the blade and the unbroken ice. As the profile penetrates the ice near the blade tip, a crack is formed, and an ice block splits from the pressure side of the blade, which causes the negative ice force $F_{ice}^-$ on the blade directed to the side opposite to the movement of the vessel.

For the type of interaction considered in Figure 2, a detailed scheme of the ice-powder extrusion is shown (see Figure 3). Shear stresses in ice powder are determined by Newton's law:

$$\tau_{ice} = \mu_{ice}\frac{dv_\xi}{d\eta}, \tag{5}$$

where $\tau_{ice}$ is the shear stress in the ice layer; $\mu_{ice}$ is the dynamic viscosity of ice powder; and $v_\xi$ is the ice-powder velocity tangential to the blade surface (see below).

The contact ice pressure $p_{ice}$ is determined from the equilibrium condition of the ice-powder element (Figure 3a) using the assumption that $p_{ice}$ be independent of $\eta$:

$$\frac{dp_{ice}}{d\xi} = \frac{d\tau_{ice}}{d\eta} \tag{6}$$

Substitution of Newton's law for shear stresses in ice powder (5) into the equilibrium condition of the ice-powder element (6) and subsequent integration of the resulting equation gives the velocity component tangential to the profile surface (the so-called "speed of ice channel walls" approach; see below):

$$v_{\xi} = \frac{1}{2} \cdot \frac{1}{\mu_{ice}} \cdot \frac{dp_{ice}}{d\xi} \cdot \eta^2 + C_1 \eta + C_2 \tag{7}$$

The integration constants $C_1$ and $C_2$ are determined from the boundary conditions while assuming that the velocity of ice powder in a channel varies linearly along the channel height, i.e., zero at an external boundary with undamaged ice, and $v_{\xi}$ at the profile surface determined by Equation (7):

$$\begin{cases} v_{\xi} = -v_x \cdot \cos\left(arctg\left(\frac{dy}{dx}\right)\right), \ \eta = 0 \\ v_{\xi} = 0, \ \eta = h_{ice}, \end{cases} \tag{8}$$

where $v_{\xi}$ is the speed of ice powder along the $\xi$-axis (i.e., projection of $v_x$ on the $\xi$-axis; and $v_x$ is the speed of penetration of the profile into the ice along the x-axis (see Figure 3).

The ice-powder discharge (the volumetric flow rate of damaged ice in the form of ice powder that is transported through a given cross-sectional area of an ice channel) $q_{ice}$ per unit of time is determined as:

$$q_{ice} = \int_0^{h_{ice}} v_{\xi} d\eta = -\frac{1}{12} \cdot \frac{1}{\mu_{ice}} \cdot \frac{dp_{ice}}{dx} \cdot \frac{1}{\sqrt{1 + \left(\frac{dy}{dx}\right)^2}} \cdot h_{ice}{}^3 - \frac{1}{2} h_{ice} \cdot v_x \cdot \cos\left(arctg\left(\frac{dy}{dx}\right)\right) \tag{9}$$

In paper [7], on the basis of linear fracture mechanics while assuming that the stress–strain state in ice follows the condition of plane strain [10], a theoretical estimate of the typical size of the crushing zone $\ell_{ice}$ was made (see Figure 3):

$$\ell_{ice} \cong 0.17 \left(\frac{(K_{IC})_{ice}}{(\sigma_{yield})_{ice}}\right)^2 \tag{10}$$

where $(K_{IC})_{ice}$ is the critical stress intensity factor for ice, and $\left(\sigma_{yield}\right)_{ice}$ is the ultimate tensile strength of ice.

The values of $(K_{IC})_{ice}$ and $\left(\sigma_{yield}\right)_{ice}$ for sea and fresh multiyear (MY) ice at low temperature and high loading rates were presented in [11,12]. Taking into account the latter, the following values were taken: $(K_{IC})_{ice} \approx (100–120) \, \text{kN/m}^{3/2}$ [11]; $(\sigma_{yield})_{ice} \approx (0.4–0.8) \, \text{MPa}$ [12]. Therefore, the value of $\ell_{ice} \approx (3 - 10) \, \text{mm}$ (see Equation (10), and the typical thickness of the ice-powder layer (channel width) at the profile tip was $(h_{ice})_{min} \sim 5 \, \text{mm}$ (see Figure 3).

For the given dimensions of the ice-crushing zone, the ice-powder discharge $q_{ice}$ can be calculated as follows:

$$q_{ice} \cong \frac{y_{ice} \, x_{ice}}{\Delta t} = y_{ice} \cdot v_x \tag{11}$$

where $\Delta t = \frac{x_{ice}}{v_x}$ is the time for the profile to pass through the crush zone (crushed-ice zone), $y_{ice} = \ell_{ice} \cdot \sin 70°$ (see Figure 3) and $v_x$ is the speed of penetration of the profile into ice along x-axis (see Figure 3).

Using Equations (9) and (11) to determine the ice pressure, one can obtain the expression:

$$dp_{ice} = -6\,\mu_{ice}\,v_x\,y_{ice}\,\frac{1}{h_{ice}{}^3}\,c\,\sqrt{1+4\left(\frac{t_{\max}}{c}\right)^2\left(\frac{dy_1}{dx_1}\right)^2}\,dx_1 - 3\,\mu_{ice}\,v_x\,c\,\frac{1}{h_{ice}{}^2}dx_1 \quad (12)$$

where $x_1 = \frac{x}{(c/2)}$; $y_1 = \frac{y}{t_{\max}}$, $c$ is the length of the profile, and $t_{\max}$ is the maximum profile thickness.

Based on the analysis of the results of the experimental studies, E.M. Appolonov proposed a hypothesis regarding the linear expansion of the ice-powder layer [13]:

$$h_{ice} = k\,\frac{c}{2}(x_1 - 1) + (h_{ice})_{\min} \quad (13)$$

where $k$ is a constant that is found from the boundary conditions when integrating Equation (12).

An analysis of experimental investigations showed that in ice-milling modes, the level of ice loads does not depend on the speed of interaction between the profile (propeller) and ice [7]. The latter is explained by the fact that the dynamic viscosity of the ice powder in the ice layer $\mu_{ice}$ is not constant. The results of experimental investigations showed that dynamic viscosity $\mu_{ice}$ was inversely proportional to the speed of the ice channel walls approach $v_\eta$ (see Equation (7) [5,7]); i.e.,:

$$\mu_{ice} = a\,\frac{1}{v_\eta} = \frac{a}{v_x \cdot \sin(arctg\,\alpha)} \quad (14)$$

where $a$ is an experimental constant [5,7]; $\alpha = arctg\left(2 \cdot \frac{t_{\max}}{c}\frac{dy_1}{dx_1}\right)$.

Considering the latter, one can express ice pressure in the form:

$$p_{ice}(x_1) = -3ac\int_0^{x_1} F\left[\left(\frac{t_{\max}}{c}\right), \left(\frac{dy_1}{dx_1}\right), (h_{ice})_{\min}, y_{ice}\right]dx_1 + C_0 \quad (15)$$

where $k$, $C_0$ are the integration constants; and $F\left[\left(\frac{t_{\max}}{c}\right), \left(\frac{dy_1}{dx_1}\right), (h_{ice})_{\min}, y_{ice}\right]$ is the function.

The integration constant $C_0$ and the constant $k$ from Equation (15) are determined from the boundary conditions:

$$p_{ice}(d\omega) = (p_{ice})_{max}(d\omega),\ x_1 = 1\ \text{and}\ p_{ice}(d\omega) \cong 0.0,\ x_1 = 0.0 \quad (16)$$

For given $p_{ice}(x_1)$, shear stresses $\tau_{ice}(x_1)$ in the ice-powder layer can be determined using Expressions (5) and (7). It should be noted that in the general case, the values of $p_{ice}(x_1)$ and $\tau_{ice}(x_1)$ are determined by the shape of the profile and the strength characteristics of the ice.

Figure 4 shows the distribution of ice pressure $\overline{p}_{ice}(\overline{\zeta}) = \frac{p_{ice}(d\omega)}{(p_{ice})_{max}(d\omega)}$ for the icebreaker profile and IK82 profile (blunt profile) (see Figure 1), where $\overline{\zeta} \in [0,\ 1]$ is the dimensionless contact zone along the profile, 0 is the beginning of a contact zone, and 1 is the end of a contact zone. It is important to note the following typical features of ice pressure. The distribution $p_{ice}$ along the edge of the profile (profile tip) is not uniform, as is customary in modern DNV-GL [14], TRAFI [15], and IACS [16] rules to ensure the strength of icebreaker blades. The distribution $p_{ice}(d\omega)$, $p_{ice}(d\omega)$ is peaked at the maximum value $(p_{ice})_{max}(d\omega)$, $(p_{ice})_{max}(\omega)$ in the leading edge of the profile. The distribution of ice pressure does not depend on the speed, which is due to the change in the viscosity of the ice powder in the channel (see above). The distribution of ice pressure weakly depends on the shape of the profile [7] and can be approximated by the equation:

$$\overline{p}_{ice} = c \cdot e^{\left(-\frac{\overline{\zeta}}{\alpha}\right)} + d \cdot e^{\left(-\frac{\overline{\zeta}}{\beta}\right)} \quad (17)$$

where c, *d*, α, β are parameters [7,8]. Equation (17) was elaborated by an approximation of numerical Solution (15).

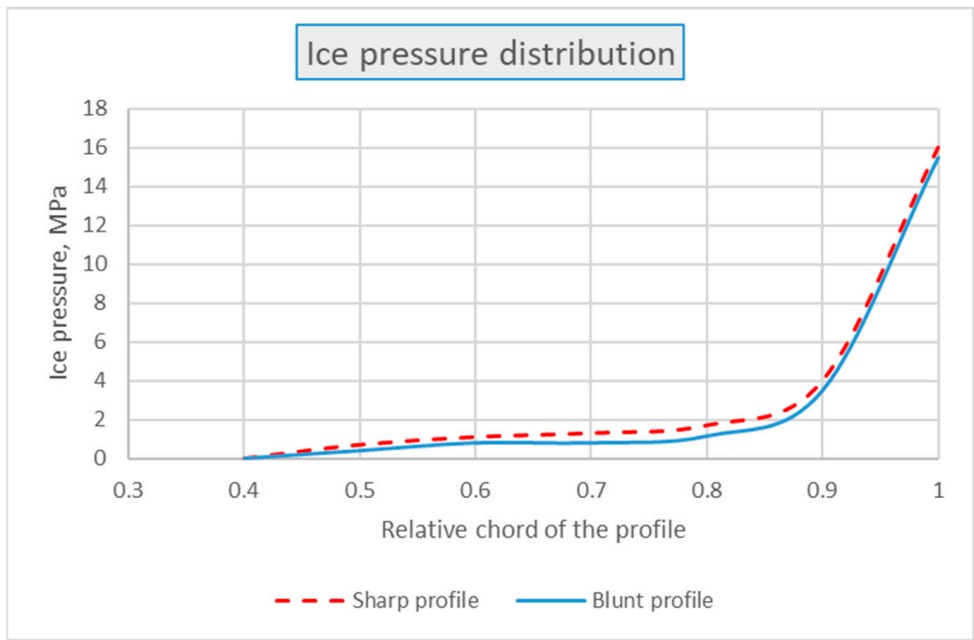

**Figure 4.** Ice pressure distribution for icebreaker profile and IK82 profile (blunt profile) [7].

Shear stresses are characterized by the regularities listed above.

For the design mode of ice milling, the maximum design ice loads (backward ice force and ice propeller torque) on the propeller are realized for angles of attack close to zero $\alpha \sim 0$ [5,7,8]. Figure 5 shows the profile contact with ice for a zero attack angle. Calculations showed that for the specified modes, the application of a sharp icebreaker profile reduced the profile ice resistance and ice moment on the propeller by 25–30%, which was confirmed by the results of full-scale tests of the side propellers of the Arctic icebreaker with blunt and sharp profiles. Profile ice resistance is determined by Expression (18):

$$F_{ice}^x = \int_{s=CBA} [p_{ice}(\zeta) \cdot \cos(n \hat{,} x) + \tau_{ice}(\zeta) \cdot \sin(n \hat{,} x)] \, ds \qquad (18)$$

Table 1 shows the calculated values of the relative profile resistance for a sharp icebreaking profile $\left(F_{ice}^x\right)_{sharp} / \left(F_{ice}^x\right)_{blunt}$ at a zero angle of attack.

**Table 1.** The results of ice profile resistance calculations both for the original and modified propeller blade profiles at zero angle of attack. Relative ice profile resistance $\left(F_{ice}^x\right)_{sharp} / \left(F_{ice}^x\right)_{blunt}$.

|  | From the Suction Side | From the Pressure Side | Total Force |
|---|---|---|---|
| Modified (sharp) profile | 0.78 | 0.61 | 0.76 |

To ensure the strength of the propeller (blade), it is also necessary to take into account the positive ice forces $F_{ice}^+$ directed toward the movement of the vessel. These forces are due to the interaction of the peripheral sections of the blade with ice blocks [5,8]. The corresponding interaction scheme is shown in Figure 6 (the second type of ice milling).



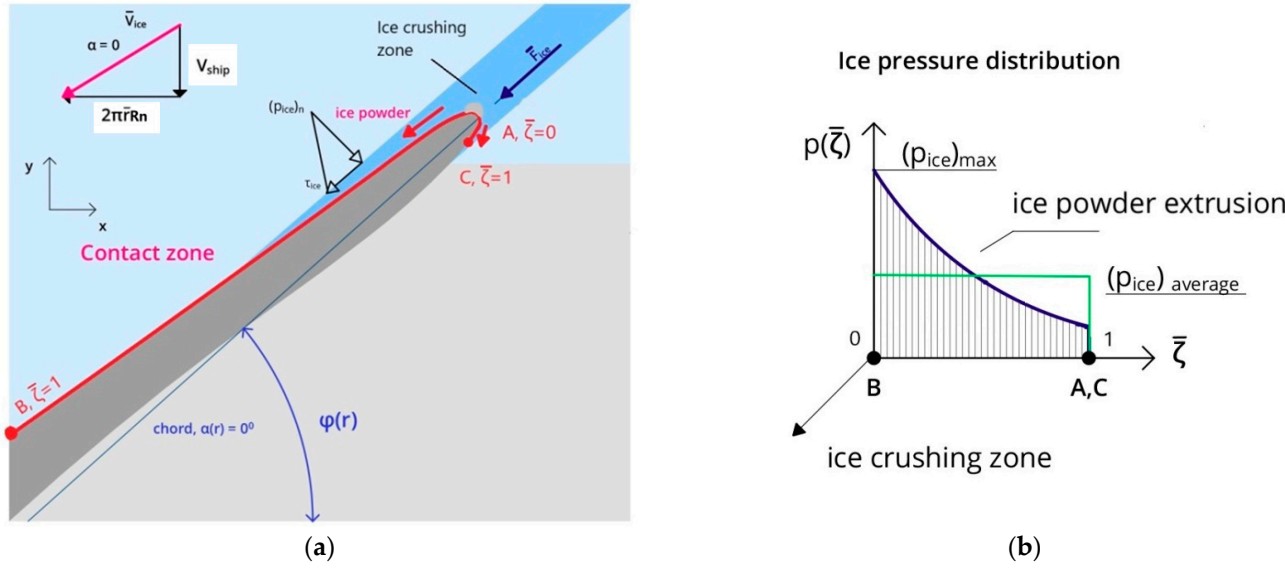

**Figure 5.** (**a**) Scheme of the profile/ice contact for zero attack angle; (**b**) ice pressure distribution along the contact zone. AB—contact zone from the suction side; BC—contact zone from the pressure side; $\overline{\zeta} \in [0, 1]$, 0—beginning of a contact zone; 1—end of a contact zone (see also Figure 3 for more details); $\alpha$—attack angle; $\varphi$—pitch angle; $V_{ship}$—ship speed; $V_{ice}$—ice contact speed; $n$—propeller speed; $\overline{r}$—relative radius of propeller; $R$—propeller radius; $(p_{ice})_{average}$—average ice pressure.

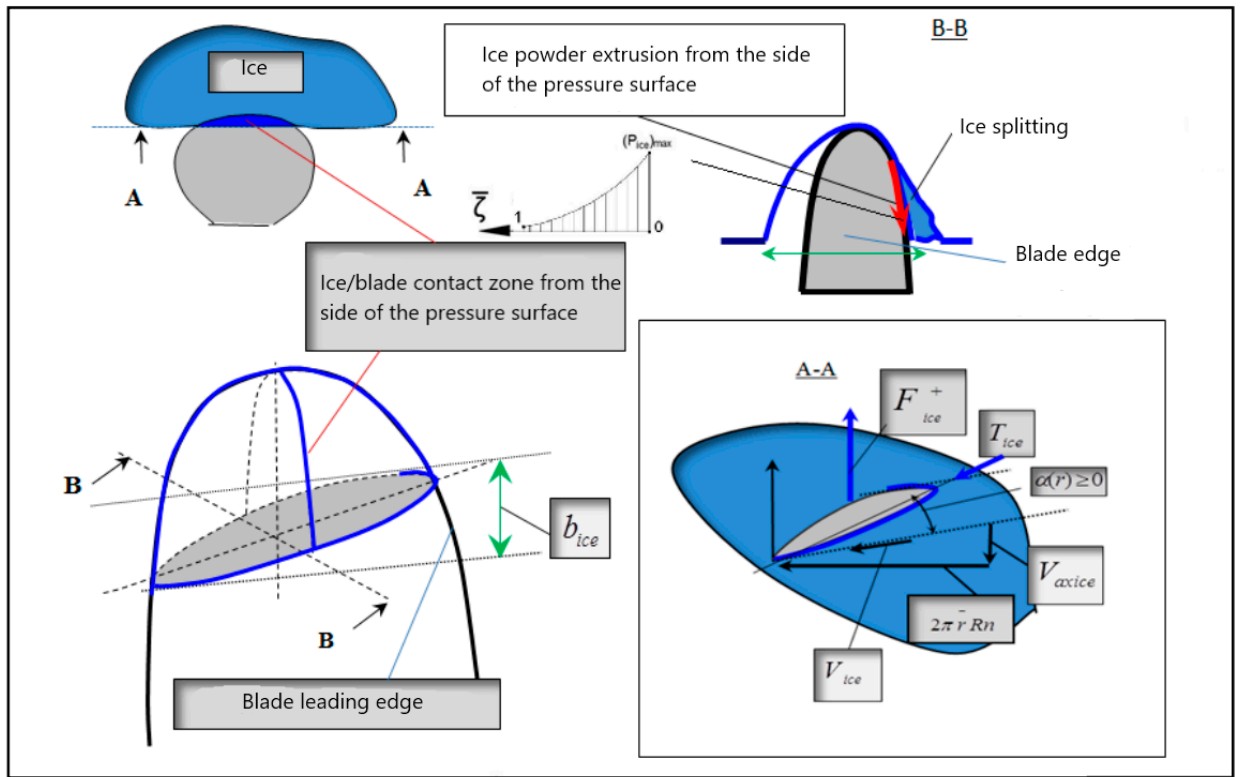

**Figure 6.** Scheme of interaction of the peripheral sections of the blade with an ice block, the second type of ice milling [5,8]. $F_{ice}^+$—forward axial ice force, $\overline{\zeta} \in [0, 1]$, 0—beginning of a contact zone; 1—end of a contact zone; $b_{ice}$—width of a cut by peripheral cross-sections. For more information, refer to [7,8].

For this type of interaction, ice is broken by the blade tip, and the broken ice is extruded along the pressure surface. Ice pressure in the contact zone is determined in the same way as for the first type of ice milling (see above).

Design schemes of ice loads in the propeller/main electric engine system and examples of their calculations for propellers with traditional (blunt) and sharp icebreaking profiles are given below.

## 3. Ice Loads Acting on a Propeller/Main Engine System and Ensuring Its Operability under Ice Conditions: Improvement of the Operational Efficiency of Icebreakers Equipped with Propellers with a Sharp Profile

The approach to determine the ice loads acting on a propeller shaft/main engine system was submitted in [8]. While operating under ice conditions, a propeller experiences backward $F_{ice}^-$ and forward $F_{ice}^+$ axial ice forces, as well as ice moment $Q_{ice}$. Attack angle $\alpha(\bar{r})$ during ice–propeller interaction, corresponding ship speed $V_{ship}$, and propeller speed $n$ are the main kinematic parameters that determine the ice loads:

$$\alpha(\bar{r}) = \phi_{pitch}(\bar{r}) - arctg\left[\frac{V_{ship}}{2(\pi\bar{r} \cdot R \cdot n)}\right] \tag{19}$$

where $V_{ship}$ is the ship speed, $n$ is the propeller speed, and $\varphi_{pith}(\bar{r})$ is the propeller pitch angle at relative radius $\bar{r}$.

Peak values of the ice loads $F_{ice}^-$, $Q_{ice}$ are governed by the first type of ice milling (see Figure 2), and correspond to bottom values of attack angle $\alpha(\bar{r})$, which are realized with a high ship speed $V_{ship}$ or with a slowdown in propeller speed $n$ during propeller/ice interaction. Ice-milling modes at a positive attack angle are considered as design modes to assign the ice loads and to ensure the strength and operability of a propulsion system. At negative attack angles or during propeller stop, an ice block impacts a propeller blade from its suction side, resulting in abrupt increase in the backward ice force $F_{ice}^-$ and possible blade damage. Indicated modes are considered as off-design modes of propeller/ice interaction, and are to be prevented while operating [7]. Forward ice force $F_{ice}^+$ is determined by the second type of ice milling (see Figure 6). Peak values of ice force $F_{ice}^+$ are realized in bollard mode when $\alpha \cong \phi_{pitch}$. Considering this, the approach to determine the ice loads acting on a propeller was developed (see Equations (20)–(23)) [8]. Equation (23) governs the evolution of the total main engine torque $Q_{total}$, propeller speed $n$, and ice forces depending on attack angle $\alpha(\bar{r})$ during propeller/ice interaction. For strength calculations, ice forces $F_{ice}^-$, $F_{ice}^+$ are assumed to be applied to the same propeller blade:

$$F_{ice}^- = 10^3\left[17.6 + 19.2e^{0.17\alpha(\bar{r}=0.9)}\right] K_{Sice} \cdot K_{Hice} \cdot D^{1.6} \cdot c_{mean} \cdot \sigma_{compr}(\bar{r} = 0.8) \tag{20}$$

$$F_{ice}^+ = K_f \cdot c(\bar{r} = 0.95) \cdot R \cdot \sigma_{compr}(\bar{r} = 0.8) \cdot k_{scale}(\bar{r} = 0.95) \cdot K_{Sice} \cdot \frac{H_{ice}}{R(1 - r_{hub})} \tag{21}$$

$$Q_{ice} = k_{profile} \cdot \left[0.22515 + 30.0 \cdot \exp(-0.07 \cdot \alpha(\bar{r} = 0.8))\right] K_{Sice} \cdot K_{Hice} \cdot D^{2.6} \cdot t(\bar{r} = 0.8)^{0.5} \sigma(r = 0.8) \tag{22}$$

$$Q_{total} = Q_{hydr} + Q_{ice} - \theta\frac{\partial n}{\partial t} \tag{23}$$

where $D$ is the propeller diameter, m; $R$ is the propeller radius, m; $\alpha(\bar{r})$ is the propeller blade attack angle at relative radius $\bar{r}$, deg (depending on ship speed $V_{ship}$ and propeller speed n); $K_{Sice}$, $K_{Hice}$ are coefficients considering ice thickness and ice strength, respectively; $t(\bar{r} = 0.8)$ is the blade thickness at relative radius $\bar{r} = 0.8$ m; $c_{mean}$ is the average dimensionless blade width along the ice-cutting depth, by RS normative procedure Book 20 [17]; $\sigma_{compr}(\bar{r} = 0.8)$ is the design-unconfined compressive strength, MPa, by RS nor-

mative procedure Book 20 [17]; $c(\bar{r} = 0.95)$ is the width of the blade cylindrical section at relative radius $\bar{r} = 0.95$, m; $K_f \overset{def}{=} K_f(\alpha(r = \overline{0.95}))$ is the coefficient obtained from the full-scale measurements of ice loads on the propeller of "Gudingen" [18]; $k_{scale}(c(\bar{r} = 0.95))$ is the scale factor of ice force $F_{ice}^+$ vs. $c(\bar{r} = 0.95))$, Equation (3); $k_{profile}$ is the coefficient considering the form of a blade leading edge; $Q_{hydr}$ is the hydrodynamic moment; $Q_{ice}$ is the average value of the propeller ice moment (without torsion vibration), Equation (22); $n$ is the propeller speed; $\theta(\partial n/\partial t)$ is the inertial component of the main engine torque; $\theta$ is the inertia moment of the main electric engine–shaft–propeller system; and $t$ is the current time of propeller/ice interaction. Factors $K_{Sice}$, $K_{Hice}$ are assigned while considering full-scale measurements of ice loads acting on the propeller and main engine of ice-going ships and icebreakers. For example, using full-scale data [19], the factor $k_{Hice}$ can be estimated with the following equation:

$$K_{Hice} = \left( \frac{h_{ice}}{(h_{ice})_{max}} \right)^{\nu} at \ (h_{ice})_{min} \leq h_{ice} \leq (h_{ice})_{max} \tag{24}$$

where $k_{Hice} = 1$ at $h_{ice} \geq (h_{ice})_{max}$; $h_{ice}$ is the design thickness of ice interacting with a propeller (thickness of thermal ice or ridge consolidated layer); $(h_{ice})_{max} = 0.335D$; $(h_{ice})_{min} = 0.17D$; $\nu \approx 1$; and D is the propeller diameter. Based on the results of full-scale measurements and direct calculations of ice loads using Expression (18) for ice-milling modes when the propeller interacts with ice blocks split from first-year (FY), second-year (SY), and multi-year (MY) ice sheets during winter–spring navigation, the indicated factor $K_{Sice}$ equals to $K_{sice} \cong 1.25 - 1.4$. For modes of propeller interaction with a ridge keel, $K_{sice} \cong 1.0 - 1.1$. To determine the indicated factor more precisely, one should utilize the calculation results of ice-sheet strength characteristics depending on ice type, thickness, and navigation period (temperature). Considering the studies performed, coefficient $k_{profile}$ was taken as 0.8 and 1.0 for sharp (icebreaker) and blunt (traditional) profiles, respectively (see Figure 1).

According to the full-scale tests, ice moment $Q_{ice}(t)$ is governed by the functions:

$$\begin{cases} Q_{ice}(t) = (Q_{max})_{ice} \cdot (1 - \exp(-d \cdot t)), & t > 0 \\ Q_{ice}(t) = (Q_{max})_{ice} \ at \ t \to \infty \ ; & Q_{ice} = 0, \quad t = 0 \end{cases} \tag{25}$$

where $t$ is the current time of propeller/ice interaction; $d$ is the parameter governing the ice moment $Q_{ice}(t)$ evolution, and is determined from the full-scale data. Peak value $(Q_{max})_{ice}$ is calculated from Equation (23) at $t \to \infty$ and $\theta(\partial n/\partial t) \to 0$, $Q_{total} = N/n$; and N is the propeller power (power of main electric engine).

It is necessary to point out that the operability of an azimuth thruster MEE is to be assured during the whole process of propeller/ice interaction; i.e., MEE is to maintain the power, torque, and propeller speed required to provide sufficient thrust, design (specified) ice-milling modes, and prevent off-design operating modes, including propeller stoppage [20]. According to the guidelines of the supplier of the "Azipod" azimuth thruster, Figure 7 shows a 'power vs. main engine torque' diagram for the modern Azipod unit. Peak torque $(Q_{engine})_{lim}$ is one of the main parameters that determines MEE operability. To ensure its operability, the following condition is to be fulfilled: average torque acting on an MEE $Q_{total}$ (Equation (23)) peak shall not exceed torque $(Q_{engine})_{lim}$. In this case, the shaft power remains constant for specified values of propeller rotational speed $n \geq n_{min}$ which ensures the propeller/ice interaction for designing ice-milling modes. Thus, condition $(Q_{engine})_{lim} \geq Q_{total}$ is necessary to ensure effective and safe operation of DAS propulsion complexes for astern mode under severe ice conditions (in ridges) [8]. In case of $(Q_{engine})_{lim} < Q_{total}$, power does not remain constant. With an increase in the ice moment $Q_{ice}(t)$, the power N and propeller speed $n$ decrease, resulting in an off-design mode of propeller/ice interaction and propeller stoppage. Figures 8 and 9 show examples of design and off-design modes, respectively, during propeller/ice interaction (evolution of torque, power, and propeller speed) for a modern large-capacity tanker moving astern in ridges.

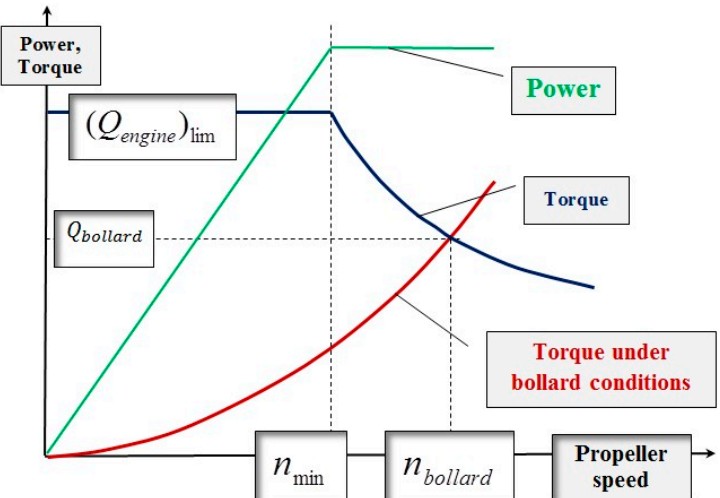

**Figure 7.** Scheme of engine power and torque for the main electric engine depending on propeller speed according to guideline of supplier of "Azipod" Azimuth Thruster [1].

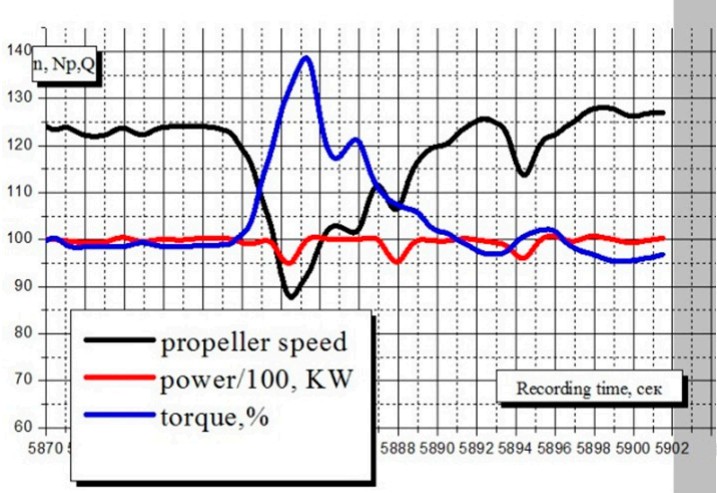

**Figure 8.** Electric engine torque change due to propeller/ice interaction at design regime at a constant power for a large-capacity tanker, astern mode in ridges [8].

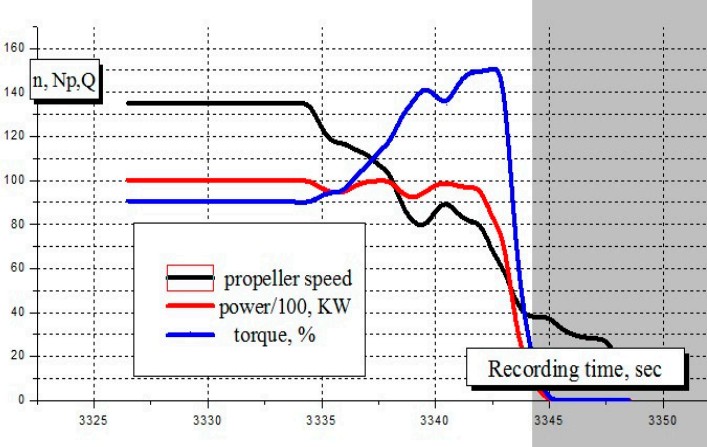

**Figure 9.** Electric engine torque change due to operation in ice at off-design regime with a rapid power and propeller speed drop. Large capacity tanker in astern mode in ridges [7].

Nowadays, electric azimuth units intended for a DAS present a major challenge for the MEE installation of the required design torque $(Q_{engine})_{lim}$ value and inside the limited space of a pod to provide sufficient operability. An increase in pod dimensions diminishes the hydrodynamic efficiency of an azimuth thruster abruptly. Therefore, reducing both ice moment $Q_{ice}$ and the designed main engine torque $(Q_{engine})_{lim}$ is an important engineering task; its solution permits increasing the operability in ice and decreasing the delivery cost of a propulsion system significantly. Experience in the design of the azimuth thrusters intended for Arctic large-capacity LNG-carriers has proved that a sharp profile of a propeller reduces the ice moment $Q_{ice}$ down to ~20%, and this became one of the key solutions that provided delivery of the azimuth thrusters with the required characteristics in order to realize the DAS concept under severe ice conditions. As an illustration of the approach above, calculations have been carried out for the icebreaker azimuth thruster characteristics in astern mode in first-year (FY) ridges with a thermal ice thickness of 2.0 m (see Figure 10). The calculations were performed for the propellers with both sharp and blunt (traditional) edges. The propeller's main characteristics are given in Table 2.

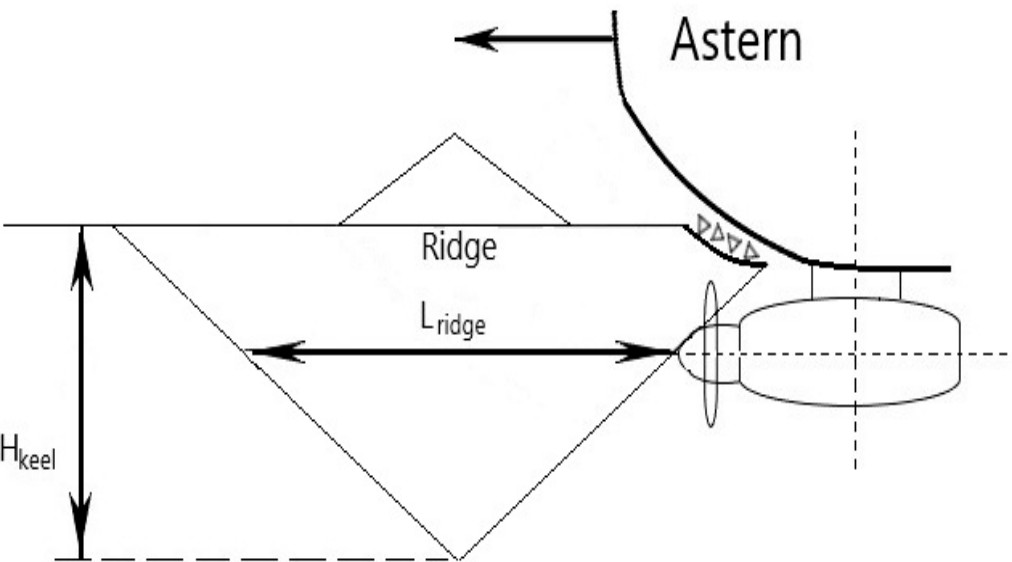

**Figure 10.** Scheme of astern movement in first-year ridges of an arctic icebreaker equipped with an electric azimuth unit. $H_{keel}$—the depth of ridge keel; $L_{ridge}$—the width of ridge keel. Ship speed is in knots and time is in seconds.

**Table 2.** The main characteristics of the propeller on the Arctic icebreaiker.

| D, m | EAR | H/D | N, kW |
|------|-----|-----|-------|
| 4.5 | 0.714 | 0.944 | 8775 |

$D$—diameter; $H$—pitch at relative radius $\bar{r} = 0.8$; $EAR$—expanded area ratio; $N$—shaft power.

To determine the ice loads on the propellers, it was necessary to take into account the speed drop in the process of overcoming the ridged isthmus. In the general case, the process of ship slowdown due to interaction with a ridge is described by Equation (26):

$$m \cdot \frac{dV(t)}{dt} = R_{water}(V(t),t) + R_{ICE}(V(t),t) + \sum T_E(V(t),t) \qquad (26)$$

where $V(t)$ is the ship speed; $R_{water}(V(t),t)$ is the hydrodynamic resistance; $R_{ICE}(V(t),t)$ is the ice resistance; $\sum T_E(V(t),t)$ is the propulsion system thrust; $t$ is the current time of propeller/ice interaction; and $m$ is the mass of the ship with added masses of water.

Ice resistance $R_{ICE}(V(t),t)$ consists of two terms: resistance due to damage to the ridge consolidated layer $R_{ICE}^{cons\ layer}(V(t),t)$, and ridge keel resistance $R_{ICE}^{Keel}(V(t),t)$. Thus, one can express ice resistance in the following form:

$$R_{ICE}(V(t),t) = R_{ICE}^{cons\ layer}(V(t),t) + R_{ICE}^{Keel}(V(t),t) \tag{27}$$

Ridge keel resistance $R_{ICE}^{Keel}(V(t),t)$ estimation can be performed based on the methodology submitted in papers [21–23]:

$$R_{ICE}^{Keel}(V(t),t) = R_{ICE}^{Keel}(t)\ V^{0.66} \tag{28}$$

where $R_{ICE}^{Keel}(t) = \int_{s(t)} \left[ p_{ridgepressure} \cdot \cos(n\hat{\ }x) + \tau_{ridgepressure} \cdot \sin(n\hat{\ }x) \right] ds$; $(n\hat{\ }x)$ is the angle between the normal to the hull surface $n$ and the ship movement direction $x$; $p_{ridgepressure} \cong 0.0149 H_{keel} + 0.0394$ is the normal stress, MPa, in the ridge keel [21–23]; $\tau_{ridgepressure} \cong 0.0017 H_{keel} + 0.0044$ is the shear stress, MPa, in the ridge keel [21–23]; $H_{keel}$ is the ridge keel depth, m; and $s(t)$ is the hull/ridge keel contact area at current time of interaction $t$.

Ridge consolidated layer resistance $R_{ICE}^{cons\ layer}(V(t),t)$ estimation can be performed based on any approved methodology of ice-resistance calculation in level ice. For example, one can refer to the methodology submitted in [24]. As a design ice thickness, one should take the design thickness of the ridge consolidated layer while taking into the account scale factor [21–23].

The results of speed estimation are shown in Figure 11.

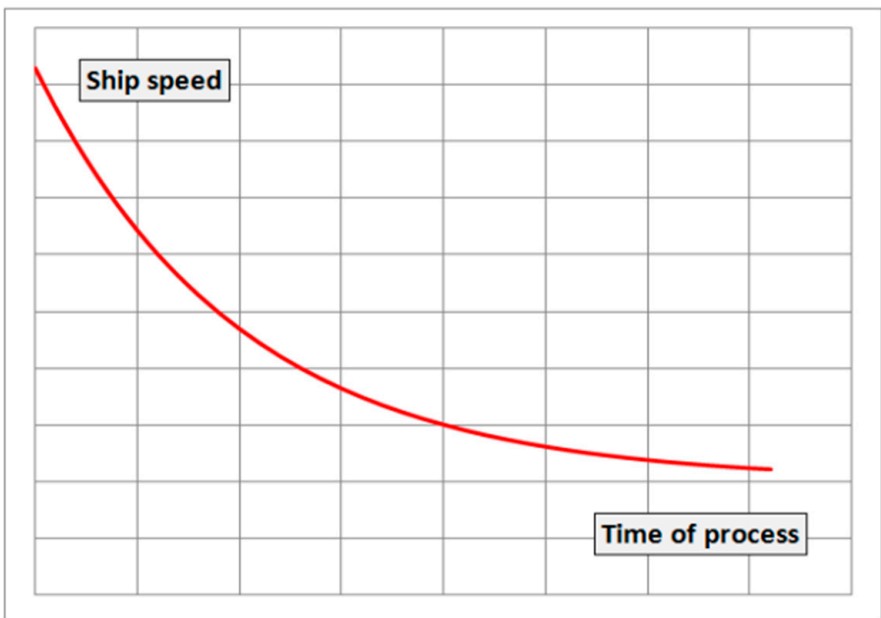

**Figure 11.** Icebreaker speed dropping while overcoming ridges astern.

Figures 12–14 show the processes of ice and hydrodynamic load variations on the propeller and the MEE, as well as the propeller speed variation during movement in ridges astern.

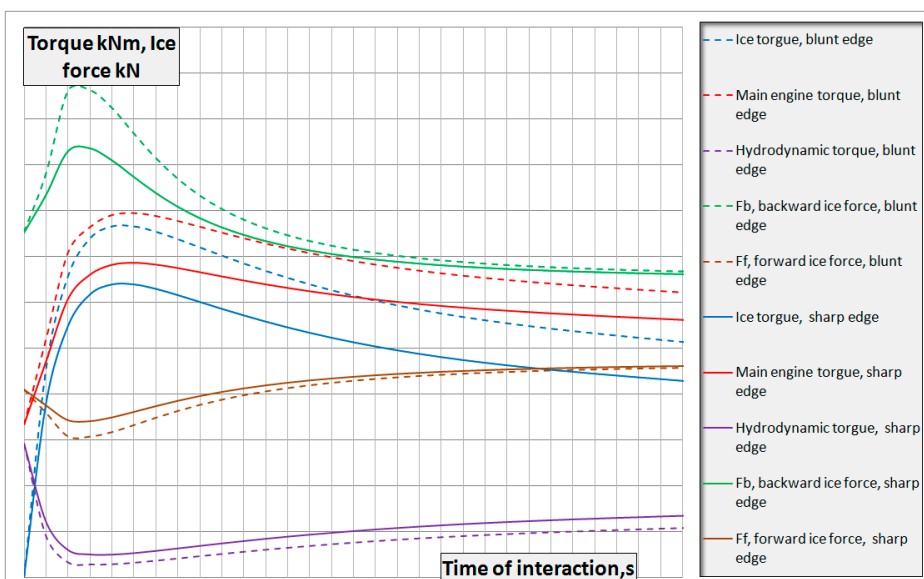

**Figure 12.** Ice loads and hydrodynamic torques on the propeller and MEE of azimuth unit during astern movement in ridges.

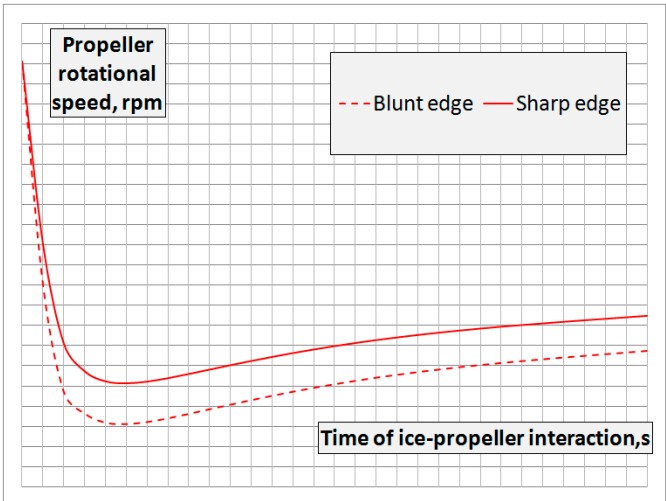

**Figure 13.** Propeller speed dropping during movement astern in ridges.

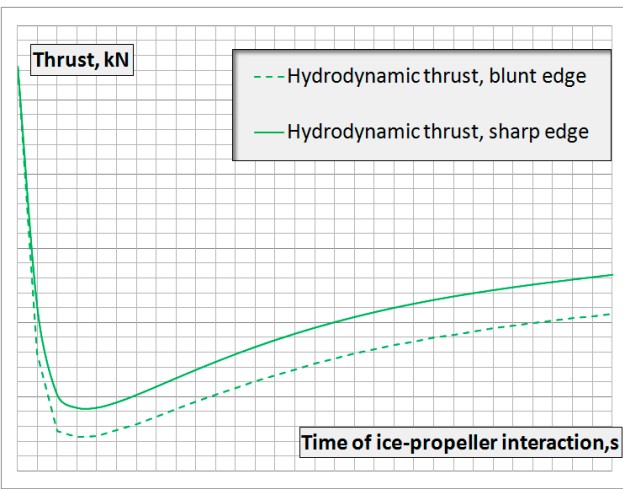

**Figure 14.** Hydrodynamic thrust variation during movement astern in ridges.



The analysis shows that a sharp profile application reduces ice loads (ice moment, axial ice force) on the propeller down to 20–25%. Reduction of the ice moment makes it possible to reduce the drop in propeller speed and improve the performance of the propulsion system. The average thrust of a propeller with a sharp profile is 20–30% higher than that with a traditional blunt profile; this significantly increases the operational efficiency of the vessel operated astern in ice conditions. It should be noted that for a sharp profile, the maximum values of the ice moment $Q_{ice}$ on the propeller and the total moment of the MEE $Q_{total} = Q_{hydr} + Q_{ice} - \theta \frac{\partial n}{\partial t}$ are reduced by ~20–25% and ~17%, respectively. The decrease in the drop of $Q_{total}$ in comparison with $Q_{ice}$ is compensated by the increase of the hydrodynamic moment due to the propeller speed gain. Reducing the total moment $Q_{total}$ allows a reduction in the design value of the moment of MEE and the delivery cost of the azimuth unit.

The implementation of these advantages can only be realized by ensuring the strength of the sharp edges of the propeller when interacting with ice. Below are the main provisions of the methodology for solving this problem.

## 4. Strength Assurance of Icebreaking Propellers with Sharp Edges

The scantlings of icebreaking propeller blades exposed to ice pressure are assigned while proceeding from assurance of both fatigue strength and static strength (strength from single extreme ice loading). Such an approach was firstly introduced in *The Russian Maritime Register of Shipping* (RMRS) by A.V. Andryushin within the development of IACS unified requirements (The International Association of Classification Requirements) [25]. The methodology has been widely used for propeller design for modern icebreaking vessels, including DAS. The methodology, which is being constantly improved, now is among the actual normative RS procedures [17]. The refined methodology of icebreaking propeller blades' strength assurance was developed by A.V. Andryushin in CNIIMF based on the indicated normative RS procedures [17,25]. The refined methodology considered approaches to determine both global and local (pressures) ice loads acting on propeller blades depending on their geometry and rotational speed, main engine power, ship speed, and morphological and strength characteristics of ice formations (see above, as well as papers [8,26–28]).

Calculation of the propeller blade stress–strain state is carried out using finite element method (FEM) under action of backward $F_b$ and forward $F_f$ axial ice forces. Ice forces $F_b$ and $F_f$ are input as distributed ice loading (ice pressure) acting on a leading edge and a tip of a propeller blade (see Figures 1–4 and 6, as well as papers [8,26–28]). Such an approach makes it possible to ensure both the general strength of a propeller blade and its local strength (blade edges, blade tip), to minimize thicknesses of blade edges and blade tip (to 'sharpen' them) in order to reduce ice loads acting on a propeller, as well as to increase its hydrodynamic efficiency and cavitation characteristics.

The permissible stress value from the condition of static strength assurance $\sigma_{perm\ s}$ is calculated according to the formula of normative RS procedure, Book 20 [17]:

$$\sigma_{perm\ s} = k_{safety}^{st} \sigma_{0,2}, \tag{29}$$

where $k_{safety}^{st}$ is the safety coefficient considering the reduce of permissible stresses while taking into account actual characteristics of castings, assumed to be equal to 0.8 for steel.

Maximum stresses of the blade shall not exceed permissible stress proceeding from static strength assurance $\sigma_{perm\ s}$.

The design fatigue permissible stress is determined while proceeding from Miner's rule (cumulative fatigue damage model) on the assumption that ice loads' probability distributions correspond to the third asymptotic law [29]. To calculate propeller blade fatigue strength, one must take into consideration the stress–strain state of a blade under action of backward $F_b$ and forward $F_f$ axial ice forces; i.e., load cycle asymmetry. Figure 15 shows the principal stress fields acting in the icebreaker propeller blade and resulting from the ice loads indicated above.

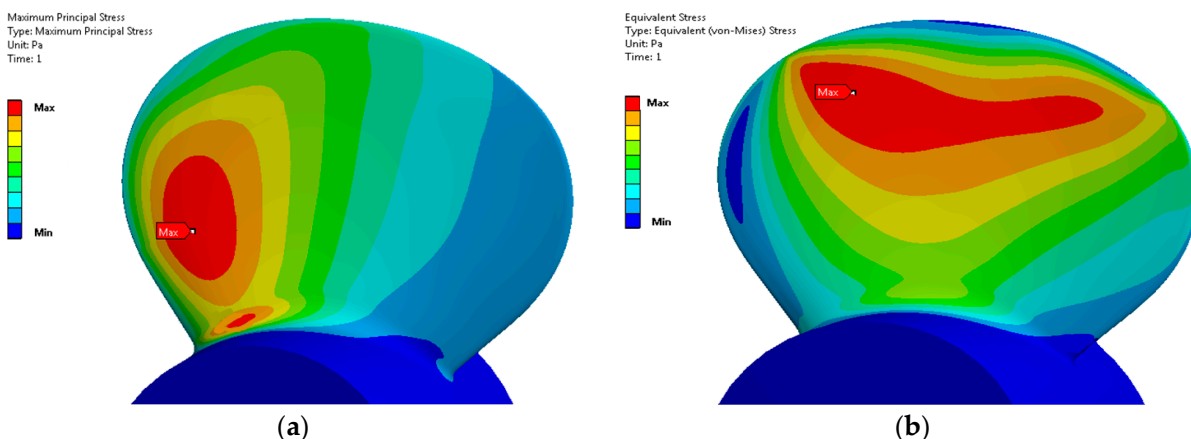

**Figure 15.** Principal stress fields on the pressure side of the icebreaker blade [7]: (**a**) backward force $F_b$ (the first type of ice milling); (**b**) forward force $F_f$ (the second type of ice milling).

Taking the latter into account, the permissible fatigue stress values (stress range) $\sigma_{perm\,f}$ are determined according to the formula [8,17,26,27]:

$$\sigma_{perm\,f} = \left(\frac{1}{k}T_{ice}n\right)^{\frac{1}{m}} \psi(m)\,\varepsilon(t)K_{mean}\,k_{var}\,k_{surf}\,\sigma_d \tag{30}$$

where $T_{ice}$ is the relative time of propeller/ice interaction; $k$ is the coefficient depending on propeller location (side, central, bow for DAS); $n$ is the propeller speed at bollard regime at full power, $s^{-1}$; $\sigma_d$ is the conditional limit of fatigue strength of a blade in sea water at number of loading cycles $N_0 = 5 \cdot 10^7$, MPa; $m$ is the constant of the material determined according to fatigue test results of standard smooth samples at symmetric cycle of loading (slope angle of fatigue SN-curve in log/log scale); $\varepsilon(t)$ is the influence coefficient of blade thickness $t$ on its fatigue strength; $\psi(m)$ is the function from $m$; $K_{mean}$ is the factor considering mean stress in the loading cycle; $k_{var}$, $k_{surf}$ are the influence coefficients of probabilistic/statistic spread of the blade fatigue strength and the influence of blade surface machining (surface hardening) on its fatigue strength, respectively. Parameters $T_{ice}$, $k$, $m$, $\varepsilon(t)$, $k_{var}$, $k_{surf}$ are determined according to the normative RS procedures [17]. Factor $K_{mean}$ takes into account the loading cycle asymmetry of a blade exposed to ice loads for the first and second type of ice milling [14].

To check fatigue strength, one must determine the maximum principal stress values $\sigma_{iceb}$, $\sigma_{icef}$ and their range $\Delta\sigma_{icemax}$ under action of backward ice force $F_b$ (the first type of ice milling) and forward ice force $F_f$ (the second type of ice milling), based on the results of stress-condition calculations:

$$\Delta\sigma_{icemax} = \left|\frac{\left(\sigma_{iceb\max} - \sigma_{icef\max}\right)}{2}\right| \tag{31}$$

To ensure fatigue strength, the range of maximum principal stresses $\Delta\sigma_{icemax}$ shall not exceed the permissible fatigue stress value $\sigma_{perm\,f}$.

The studies proved that in order to apply a sharp profile, it is necessary to use propeller martensitic–austenitic steels of a high-yield strength $\sigma_{yield}$ and fatigue properties in sea water. It is advised to use steels with $\sigma_{yield} > (550 - 600)$ MPa. In this case, 'fatigue' is the main factor to ensure the strength of steel propellers and assign their scantlings. It is important to note that surface hardening improves blade fatigue strength, resulting in reduced scantlings of propeller blades. This leads to a better performance, including a reduction in the blade damage force.

## 5. Improvements in Hydrodynamic and Cavitation Characteristics of Icebreaking Propellers with a Sharp Profile

The use of sharp profiles significantly improves the hydrodynamic and cavitation characteristics of the propeller. Figure 16 shows a relative comparison of the efficiency of icebreaking propellers with a sharp profile and a blunt profile of type IK82 (see Figure 1), obtained from the results of testing models of propellers and calculating propeller curves by CFD. The main characteristics of the propeller are presented in Table 2. To evaluate the efficiency for the bollard mode, the ratio $K_T/K_Q^{2/3}$ was used, where $K_T$, $K_Q$ are the thrust and moment coefficients, respectively. For modes close to the navigational ones, the efficiency of a propeller with sharp edges increased by 4% compared to a propeller equipped with a blunt profile. A 4% increase in propeller efficiency significantly improved the open-water performance.

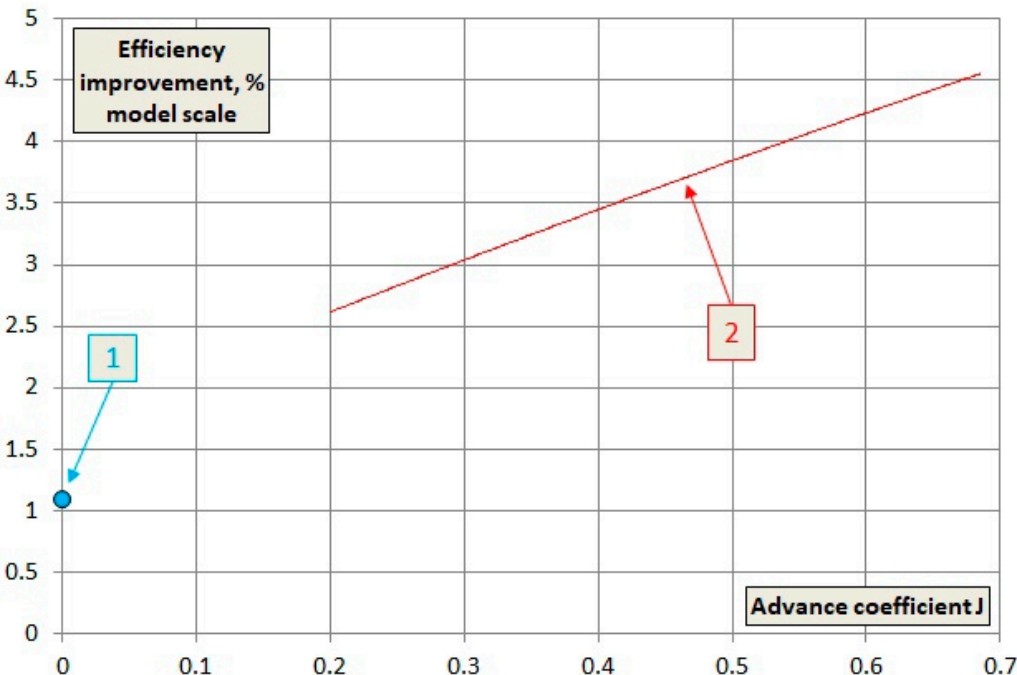

**Figure 16.** Relative comparison of the efficiency (performance factor) of icebreaking propellers with a sharp profile and a blunt profile of IK82 type (see Figure 1). 1—ratio $K_T/K_Q^{2/3}$ in bollard mode; 2—ratio $\eta_0$ for design modes.

Application of a sharp profile also significantly improved cavitation performance. Figure 17 shows cavitation diagrams of propellers with sharp and blunt profiles for bollard mode (model tests). The analysis showed that the use of a sharp profile could reduce the disk ratio by ~15%, and further improved the hydrodynamic efficiency. Reducing the disk ratio, it also made it possible to reduce the strength sizes of the propeller (blade) and reduce the ultimate blade damage force to ensure pyramidal strength.

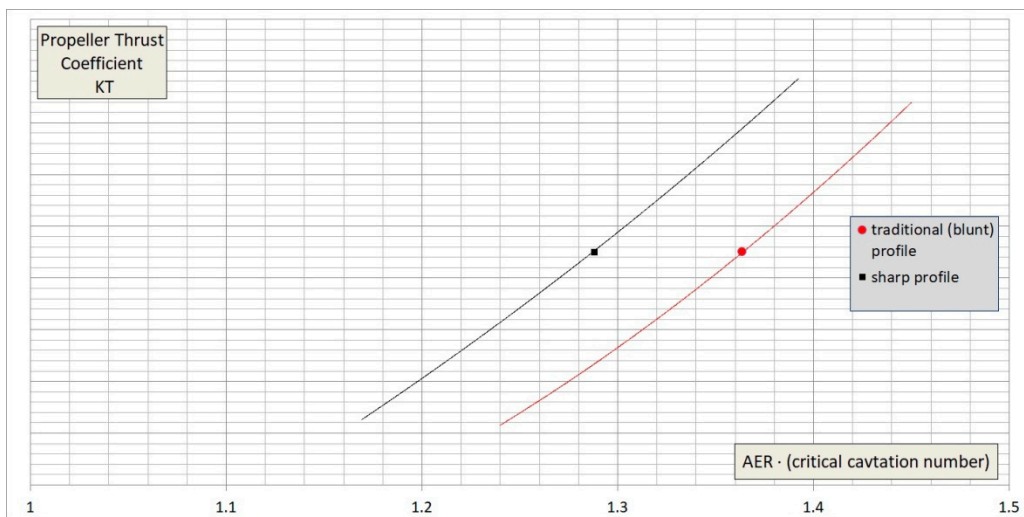

**Figure 17.** Cavitation diagrams (thrust breakdown point) of propellers with sharp and blunt profiles in bollard mode (model tests).

## 6. Conclusions

The paper considered the main aspects of the technology for ensuring the strength of icebreaking propellers and the operability of the main electric engines of modern azimuth units. Ensuring the operability of the MEE of an azimuth unit means the ability to withstand the ice moment; maintain the specified power, ice torque, and propeller speed to provide sufficient thrust and speed of the ship; as well as to prevent propeller stop and possible blade damage. The solution to this problem is essential to ensure the efficient operation of a DAS astern under ice conditions (in ridges). The propellers constantly interact with ice, and the MEE is affected by an additional ice moment that significantly exceeds the hydrodynamic moment. At present, this issue dominantly concerns large-capacity Arctic tankers. Ensuring MEE operability, as well as sufficient strength of the propeller and other elements of the propulsion complex, are interrelated. Determination of ice loads on the propeller and MEE, as well as the development of methods for their reduction, is the initial step to overcome this challenge. The use of a sharp profile for propellers (sharpening the leading edges of the blades) is one of the effective ways to reduce ice loads, as well as to increase the efficiency of the MEE and the performance of a DAS operated in ice astern. However, the use of sharp profiles was restrained by the solution of the problem of ensuring the strength of the edges of icebreaking propellers under the action of a distributed ice load. This paper presented methods for determining the contact ice pressure on the edges of propeller blades and ensuring their fatigue and static strength. The development of these methods made it possible to use sharp profiles (sharp edges) to reduce ice loads on propellers and the MEE. Ice loads could be reduced by up to 25%, resulting in the development of electric azimuth units that ensured the efficient operation of a DAS in Arctic conditions. The design of propellers showed that sharp profiles significantly improved their hydrodynamic and cavitation characteristics. The hydrodynamic efficiency increased by 4% in full-speed mode. The disk ratio from the condition of preventing the second level of cavitation for bollard mode could be reduced by 10–15%, providing further improvement of the hydrodynamic characteristics, as well as reductions in the size of the blade strength and blade damage force, to ensure the pyramidal strength of the propulsion system.

**Author Contributions:** Conceptualization, A.V.A.; methodology, A.V.A.; formal analysis, S.V.R., A.Y.V. and E.V.S.; investigation, S.V.R., A.Y.V. and E.V.S.; writing—original draft preparation, S.V.R.; writing—review and editing, A.V.A. and S.V.R.; supervision, A.V.A.; project administration, A.V.A. All authors have read and agreed to the published version of the manuscript.

**Funding:** This research received no external funding.

**Institutional Review Board Statement:** Not applicable.

**Informed Consent Statement:** Not applicable.

**Data Availability Statement:** The manuscript contains no confidential information. There are no possible copyright issues. All the data used in the analysis is the open source data only and can easily be found in the reference list provided.

**Conflicts of Interest:** The authors declare no conflict of interest.

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
