# Peer review of "Sharp Profile for Icebreaking Propellers to Improve Their Ice and Hydrodynamic Characteristics"

_jmse, doi:10.3390/jmse10060742_

Round 1

Reviewer 1 Report

The paper presents a model to calculate the ice pressure and then ice loading on a propeller blade assuming it indenting solid, and quite massive ice feature. The model is based on investigating the flow of crushed ice from the ice failure zone.

The paper is interesting in itself as it provides a model to calculate the propeller ice forces. The drawback of the paper is, however, that many of the equations are either poorly explained or just taken from some references related to ice rules (e.g. equations 20 - 23). The derivation and also origin of equation must be clearly explained so that derivation is possible to follow. It is not enough to justify some constant or value by reference to some document as this way the origin of the value is not clear. Examples of this are foe example in row 125 (25 % power reduction - is this average power, some peak power or what?) and row 148 (crushing zone ... 10 mm? Is this a result of observation?).

Eqs. (1) and (2) must be justified (they are also dimensional but dimensions are missing). Eq. (3) should also be justified and the quantity l explained. Eq. (4) relies on Mohr Coulomb ratio but this is not clear (remember that Mohr Coulomb equation is τ = σtanφ+c). Integration of eqs. (5) - (7) should be explained, for example pice is assumed independent of η. Boundary conditions (8) express the speed of crushed ice at the both sides of the channel but the first part requires elaboration. The dynamic viscosity (14) is equated with velocity and other variables - how is this equation used; are the variables measured and thus the viscosity determined (viscosity should be a constant). Eq. (17) must be elaborated, a reference to [6] is not enough. Calculations after row 384 are not clear; what are calculated and what are the assumptions? Figures 16 and 17 refer to some model tests but what these are is unclear.

The above list does not contain all the obscure or opaque parts of the paper but serve as a guide what should be edited. Overall the derivations and assumptions should be made much clearer before the derivations can be considered to be on a scientific basis. Now the derivation is an engineering model which aims to obtain desing forces but the assumptions along the way are at best justified by the end result.

Author Response

Please see the attachment (pdf)

Reviewer 2 Report

 Thank authors for making effort to this valuable study. I don' t have any major corrections, but please take a time to answer to my questions and comments as below:

Figure 3

 the position of the note ' leading edge' is not well designed. please move this to appropriate position

Figure 4 should be improved

Eq(9)  h must be hice?

Definition of Qice is difficult to understand because it appears L187, L205, L295 and L300 with different introductions

L297 Vship should be replaced by Va or Vp : propeller advance speed

Figure 7 if the x axis is liner, I believe torque curve should be liner

Figure 8-9 unit is not clear

Table 2 AER disk area must be EAR stands for expanded are ratio

Figure 11 has no scale

Figure 17 AER: please clarify this difintion

Figure 17 Does Kt mean thrust coefficient at thrust break down point?

                If so, it will be better to mention.

Author Response

Please see the attachment (pdf)
